# Silicate-Promoted Phosphorylation of Glycerol in Non-Aqueous Solvents: A Prebiotically Plausible Route to Organophosphates

**DOI:** 10.3390/life7030029

**Published:** 2017-06-29

**Authors:** Maheen Gull, Brian J. Cafferty, Nicholas V. Hud, Matthew A. Pasek

**Affiliations:** 1University of South Florida, 4202 E Fowler Ave. NES 204, Tampa, FL 33584, USA; ambermaheen@yahoo.com; 2Department of Chemistry and Chemical Biology, Harvard University, 12 Oxford Street, Cambridge, MA 02138, USA; bcafferty@gmwgroup.harvard.edu; 3Georgia Institute of Technology, 315 Ferst Drive, Atlanta, GA 30332, USA; nick.hud@chemistry.gatech.edu

**Keywords:** prebiotic synthesis, phosphorylation, origin of life, deep eutectic solvents, formamide, mineral catalysis

## Abstract

Phosphorylation reactions of glycerol were studied using different inorganic phosphates such as sodium phosphate, trimetaphosphate (a condensed phosphate), and struvite. The reactions were carried out in two non-aqueous solvents: formamide and a eutectic solvent consisting of choline-chloride and glycerol in a ratio of 1:2.5. The glycerol reacted in formamide and in the eutectic solvent with phosphate to yield its phosphorylated derivatives in the presence of silicates such as quartz sand and kaolinite clay. The reactions were carried out by heating glycerol with a phosphate source at 85 °C for one week and were analyzed by ^31^P-nuclear magnetic resonance (NMR) spectroscopy and mass spectrometry (MS). The yield of the phosphorylated glycerol was improved by the presence of silicates, and reached 90% in some experiments. Our findings further support the proposal that non-aqueous solvents are advantageous for the prebiotic synthesis of biomolecules, and suggest that silicates may have aided in the formation of organophosphates on the prebiotic earth.

## 1. Introduction

Phosphorus is an essential element for life. It is present in living organisms most prominently in the form of phosphate esters [1,2,3]. Glycerol-phosphate is an important biological phosphate ester as it is an integral part of the cell membrane, within phospholipids, and it is found in metabolic pathways involving 3-carbon sugars [4]. Glycerol serves as an excellent model phosphorylation substrate, as it bears both primary and secondary alcohols that can form phosphoesters by reacting with phosphate. From a prebiotic perspective, glycerol phosphate may have been involved in the assembly of protocells [5], and nucleic acids containing glycerol phosphate-based backbones have been proposed as possible predecessors of the ribophosphate backbone used in RNA [6,7,8]. Thus, the study of the origin of prebiotically plausible synthetic pathways that assemble glycerol-phosphate may aid our understanding of how this important molecule first entered biology [4].

Glycerol and glycerol phosphates are potentially prebiotic molecules. Glycerol has been identified in the Murchison meteorite, along with many other biomolecules, suggesting that it was present in the prebiotic milieu at the time of the origin of life. It is plausible that meteorites delivered glycerol to the early Earth [9], and recent studies have shown that it can be produced by exposing methanol-based interstellar ices to ionizing radiation [10]. The prebiotic synthesis of glycerol phosphates (i.e., glycerol-1-phosphate, glycerol-2-phosphate, and glycerol-1,2-cyclic phosphate (Figure 1)) has been demonstrated under hydrothermal conditions, as well as in reactions between phosphate, glycerol, and a condensing agent [11,12,13,14]. Additionally, glycerol phosphates can be made by using the meteorite mineral schreibersite, (Fe,Ni)_3_P, under mild heating in aqueous solutions [15]. We were encouraged by these results to explore the phosphorylation of glycerol in various prebiotically relevant, non-aqueous solvents, such as formamide and anhydrous glycerol-based solvents, and to test the role of minerals in these reactions.

For over 40 years, formamide has been examined as both a reactant and solvent for the formation of biomolecules under model prebiotic conditions [16,17,18,19]. The prebiotic synthesis of formamide has been reported using simple precursors such as NH_3_ and CO [20,21], and the molecule has been detected in the interstellar medium [22]. The advantage of using formamide instead of water as a solvent in model prebiotic reactions is due, in part, to the fact that formamide can support condensation reactions, such as phosphoester formation [23]. For example, formamide has been used as a solvent for the phosphorylation of nucleosides such as adenosine and uridine [23,24,25,26]. Despite these successes, it has been argued that there are few geochemical routes to pure formamide puddles on the early earth [27].

Recent studies show that glycerol can also be phosphorylated in non-aqueous deep eutectic solvents (DES) [14]. We have previously proposed that DES and ionic liquids might have formed on the prebiotic earth as a result of drying of aqueous salt solutions [14,28]. Of particular interest to us are non-aqueous DES composed of glycerol and choline chloride [29] as it was recently shown that these solvents support the base pairing of nucleic acids [30]. We were motivated by this report to evaluate if solvents of glycerol and choline chloride, in addition to supporting nucleic acid function, might also enable reactions that are necessary for the synthesis of nucleic acids or nucleic acid analogs.

Simple silicates, such as quartz sand (SiO_2_) and kaolinite (Al_2_Si_2_O_5_(OH)_4_), have been investigated for decades for their ability to catalyze model prebiotic polymerization reactions, including the polymerization of organophosphates (i.e., nucleotides) [31,32]. The catalytic ability of silicates, however, to promote the formation of organophosphates such as nucleotides from orthophosphate and simple organic molecules has not been thoroughly investigated. Quartz sand is a mineral common to evolved continental crust, and was likely present in some form since the earth differentiated [33]. Kaolinite is expected to have been present on the early earth as it is a natural weathering product of simple minerals such as feldspar. Kaolinite has also served as a potential catalyst in several other phosphorylation reactions [11,34,35]. Previous studies on the interaction between formamide and kaolinite [36] encouraged us to investigate if mixtures of formamide and kaolinite can promote the phosphorylation of glycerol.

We investigate here the phosphorylation of glycerol in both formamide, and the DES composed of glycerol and choline chloride (2.5:1). Glycerol-based eutectic solvents, such as this one, have been referred to as “deep eutectic” solvents even though glycerol has a melting point of 18 °C because mixtures of glycerol and choline chloride (mp 302 °C) can have melting points as low as −40 °C [29,37]. We have included it here as it parallels a urea-choline chloride solvent investigated previously [14]. We additionally examined the role of silicates, including quartz sand and kaolinite, within these solvents in catalyzing reactions between glycerol and phosphate. The phosphate sources tested include sodium phosphate, sodium trimetaphosphate (a condensed polyphosphate that has been proposed to be prebiotic [38]) and the phosphate mineral struvite—MgNH_4_PO_4_ × 6H_2_O—a mineral recently shown to form spontaneously from apatite in MgSO_4_ and ammonium formate-bearing solutions [28]. All three compounds are known to phosphorylate organic molecules under certain conditions [2].

## 2. Materials and Methods

Sodium-trimetaphosphate and formamide were obtained from Fischer Scientific (Pittsburgh, PA, USA). Choline chloride, NaH_2_PO_4_ and D_2_O were obtained from ACROS Organics (New Jersey, USA) and glycerol from Alfa Aesar (Tewksbury, MS, USA). White quartz sand (200–800 μm grain size) and kaolinite were obtained from MP Biomedicals (Santa Ana, CA, USA). All chemicals and minerals were used as received. Deionized water was obtained using a Barnstead (Dubuque, IA, USA) NANO pure^®^ Diamond Analytical combined reverse osmosis-deionization system. Struvite was synthesized by reaction of equal volumes of 0.5 M MgCl_2_, NH_4_Cl and Na_2_HPO_4_ solutions, resulting in a white precipitate that was identified by an Enwave Raman microscope (Irvine, CA, USA) as struvite, as previously reported [13].

The glycerol-choline chloride deep eutectic solvent (DES) was prepared by mixing choline chloride and glycerol in a 1:2.5 molar ratio, respectively, following prior work with deep eutectic solvents [14,29,37].

### 2.1. Phosphorylation of Glycerol in Formamide and a Deep Eutectic Solvent (DES)

In a typical phosphorylation reaction performed in formamide, 0.8 g glycerol and 0.1 g of an inorganic phosphate (P_i_) source were added to a glass vial containing 7 mL formamide. When silicates were examined, 0.5 g of kaolinite or quartz sand was added to the solutions prior to heating. The reaction vials were sealed tightly and heated with constant stirring at 85 °C for one week. To examine the phosphorylation of glycerol in glycerol-based solvents, solutions were prepared by mixing glycerol and choline chloride to give a final ratio of 2.5 to 1, to which various phosphate sources and silicates were added. Reactions were analyzed by ^31^P-NMR and mass spectrometry (MS).

### 2.2. ^31^P-NMR and Mass Spectrometry (MS) Studies of the Phosphorylation Products

^31^P-NMR spectra were acquired on Unity INOVA 400 spectrometer (161.84 MHz for ^31^P and 399.882.54 MHz for ^1^H) (Varian, Palo Alto, CA, USA) equipped with a variable temperature controller and a Varian 5 mm Autoswitchable probe with Z-axis gradient optimized for tuning of ^31^P and ^1^H [13,14,15]. The ^31^P chemical shifts are reported using an external reference standard (neat solution of phosphoric acid at room temperature, ~25 °C, δ ppm = 0.0). A ^31^P 45° flip angle pulse was used for both proton decoupled and non-decoupled spectra (90° ^31^P-pulse of 9.8 μs at 54 dB attenuation, where max power output is ~300 W). For proton decoupled spectra a composite pulse Waltz decoupling sequence was applied with field strength of 2525 Hz during the acquisition time of 1 s and the relaxation time of 1 s. Samples were dissolved in D_2_O and the signal was averaged from 512 transients. Mass spectrometry was performed in negative ion mode on a 6130 Single Quadrupole Mass Spectrometer (Agilent, Santa Clara, CA, USA) attached to an Agilent 1200 HPLC.

## 3. Results and Discussion

We initially investigated the phosphorylation of glycerol in the glycerol-choline Cl DES (Figure 1). Reactions were performed at 85 °C for one week, and analyzed by ^31^P NMR (Figure 2) and mass spectrometry [14]. Yield of organophosphates was determined by ^31^P NMR analysis and are reported in Table 1. Yields as high as 76% were observed after 7 days of heating trimetaphosphate (TMP) in the DES in the presence of kaolinite.

Sodium phosphate and sodium trimetaphosphate both dissolved to completion in formamide. Struvite was less soluble, dissolving at about 1 g/L, which is still higher than its solubility in water [39]. All salts were less soluble in the glycerol-choline chloride eutectic, with TMP most soluble at 0.2 g/L, sodium phosphate 0.02 g/L, and struvite 0.01 g/L. In all cases, dissolved phosphate was the limiting reagent.

No TMP was observed among the products of these reactions, suggesting all TMP had reacted in solution. Pyrophosphate was detected only in those experiments using TMP, where it is a hydrolysis product of TMP, with water promoting this hydrolysis likely coming from the phosphorylation reaction or present initially in the solvent (<0.5 wt.%) or from ambient humid conditions. No pyrophosphate was detected in experiments involving only struvite, in contrast to prior work [40].

With respect to the three phosphate sources, TMP was found to produce the highest yield of glycerol phosphate and choline phosphate in the DES (Table 1 and Figure 2). This was not unexpected as TMP bears high energy phosphoanhydride bonds and has been shown previously to phosphorylate organic molecules [2,41]. Sodium-phosphate and struvite are also known to be good sources of phosphate for phosphorylation, however, the yields typically are not as high as reported here [2]. Intriguingly, the total yield of organophosphates significantly improved when sand and/or kaolinite were included in the reactions (Figure 3). Both silica-based minerals promoted the formation of organophosphate compounds, with reactions performed with kaolinite producing slightly higher yields than reactions performed with quartz.

We next investigated the ability of silicate minerals to enhance the formation of organophosphates in formamide solutions. Similar to reactions performed in the DES, reactions in formamide were performed at 85 °C for up to one week. Yields were significantly enhanced in solutions containing silicate minerals for all three phosphate sources as compared to reactions that were performed in the absence of a silicate (Table 2, Figure 4). 

The best organic phosphorylation yields were obtained when struvite was used as a phosphate source and silicates were used as catalysts. To the best of our knowledge, this is the first study showing that a two-mineral, one-pot reaction produces higher yields than single mineral reactions, suggesting either some interplay between the mineral reactivity, or catalytic activity promoting the reaction.

The silicates enhanced the phosphorylation reaction in formamide, with yields of organophosphates being considerably higher than in the DES, and to the best of our knowledge, higher than any previously reported model prebiotic phosphorylation reaction performed in formamide [23,24], with TMP as the phosphate source. The heating of the reaction mixtures here for one week in the presence of silicates also promotes the formation of glycerol cyclic phosphodiesters.

The experiments reported here were all stopped and analyzed at one week. It is likely that some of these reactions will provide different yields upon longer or shorter incubation times, as suggested by prior work [23,24,26]. The rate of phosphorylation may be relatively slow, especially for those reactions contingent on dissolution of phosphate minerals (e.g., struvite), hence maximum levels of phosphorylation may require longer than the one-week reaction time investigated here.

These results are somewhat comparable to our previous findings that non-aqueous solvents like DES enhance the yields of phosphorylation reactions [14]. In our previous works we demonstrated that DES of choline chloride and urea can be used as a medium for the phosphorylation of organics (with yields up to 99%). The results present in this work show the efficiency of a glycerol-based DES and formamide as suitable mediums for phosphorylation reactions. From the prebiotic perspective, both glycerol and choline are potentially available as organic compounds on the early earth [10,42,43,44]. Glycerol is well known as a simple alcohol, produced under plausible prebiotic conditions for instance by ionizing radiations of methanol-based interstellar ices, and is present as the most abundant 3-carbon sugar-alcohol in the Murchison meteorite [9,10]. It is also formed by reduction of glyceraldehyde [12]. Choline may be accessible under prebiotic conditions [42,43,44], and if present in bodies of water, on dehydration the dominant salt of choline could have been the chloride. Formamide has been considered to be prebiotically relevant [22,23,24,25,26], albeit with caveats [27]. Hence quite possibly a glycerol-based solvent could arise through the evaporation of water with high concentrations of these two organic compounds or a plausible prebiotic formamide scenario could be established by the high concentration of formamide in the water. Subsequent water evaporation could have promoted the prebiotic formamide chemistry [20].

Furthermore, although the existence of widespread struvite on the early Earth is debatable [13], localized occurrences are certainly feasible [28] and this phosphate mineral has shown remarkable reactivity towards organics. Moreover, the present work also focuses on the potential significance of struvite as an important phosphate releasing mineral from the prebiotic perspective of phosphorylation in alternative, non-aqueous solvents.

The products are also functional group specific; most phosphorylation products in these reactions were primary alcohols. We will not claim here that this experiment is fully representative of actual prebiotic chemistry that occurred on the early earth; further work will be necessary to determine if other plausible sources of phosphorus, such as apatite [45], also promote phosphorylation (see [28] for a new route), and that a DES can arise under realistic geochemical conditions. However, the results show promise for anhydrous environments and reactions in prebiotic chemistry. We also note that the reaction pathway suggests other efficient dehydration reactions such as sugar and amino acid condensation will occur in DES systems, with potential relevance in prebiotic chemistry, biochemistry, and chemical engineering [46,47]. Such a role has been suggested for montmorillonite, in the formation of nucleotide polymers under model prebiotic conditions [32]. The present study demonstrates that, in addition to the reported ability of silicates to catalyze the formation of long RNA polymers, silicate minerals can also assist in directing dehydration reactions to form organophosphates. Thus, the results presented here suggest a potentially simple scenario for the origin of informational polymers, involving a single environmental location where silicates assisted in both the formation of (pre)RNA monomers and polymers.

## Figures and Tables

**Figure 1 life-07-00029-f001:**
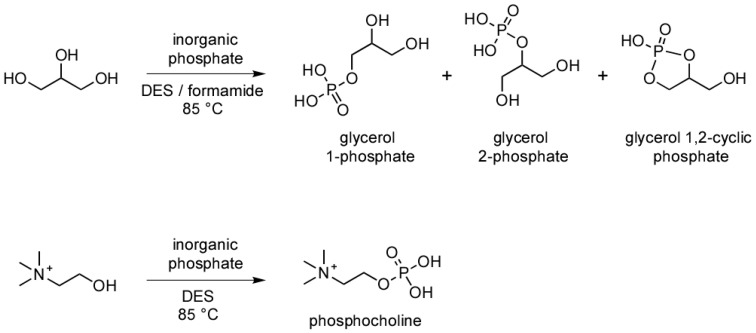
Reaction pathways, and structures of the molecules discussed in the text.

**Figure 2 life-07-00029-f002:**
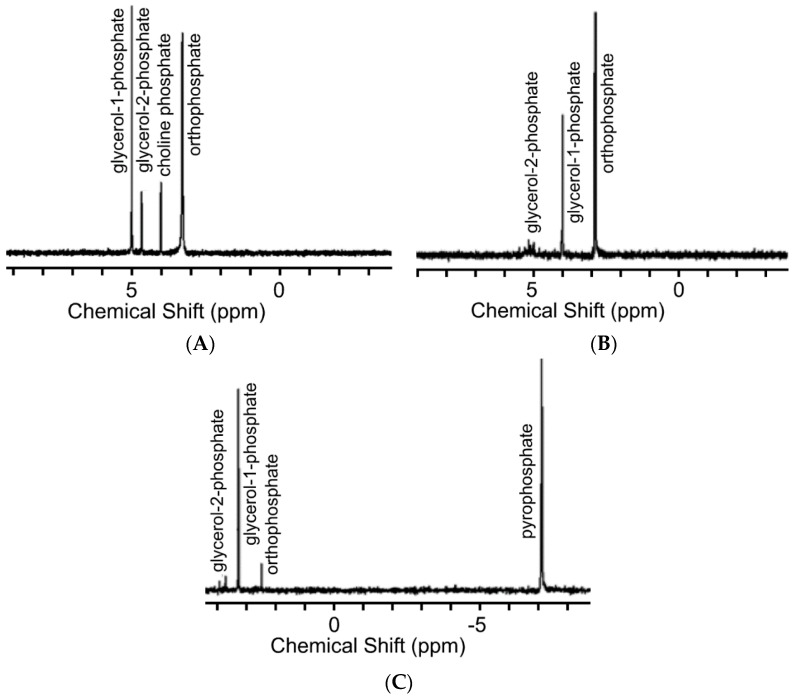
^31^P-NMR spectra of products formed by reacting (**A**) glycerol with struvite in the deep eutectic solvent, (**B**) glycerol with struvite in formamide, and (**C**) glycerol with TMP in formamide. All reactions were performed at 85 °C for seven days. Note that in formamide TMP partially hydrolyses into pyrophosphate (PPi).

**Figure 3 life-07-00029-f003:**
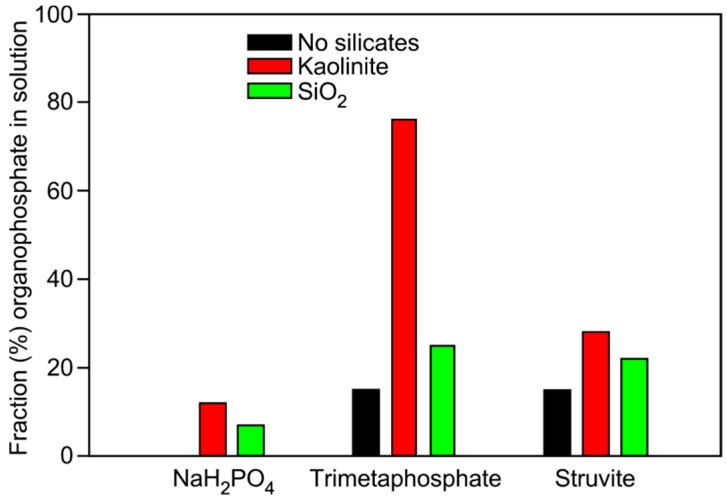
Bar graph showing the total yield of organophosphate (i.e., total % yield of all glycerol phosphate species and choline phosphate) from reactions performed in glycerol-based eutectic solvents. Reactions performed in the presence of quartz sand, kaolinite and no catalyst are shown, with the various phosphate sources used, i.e., NaH_2_PO_4_, TMP, or struvite. Errors are about 1%, based on repetition of integration of NMR peaks.

**Figure 4 life-07-00029-f004:**
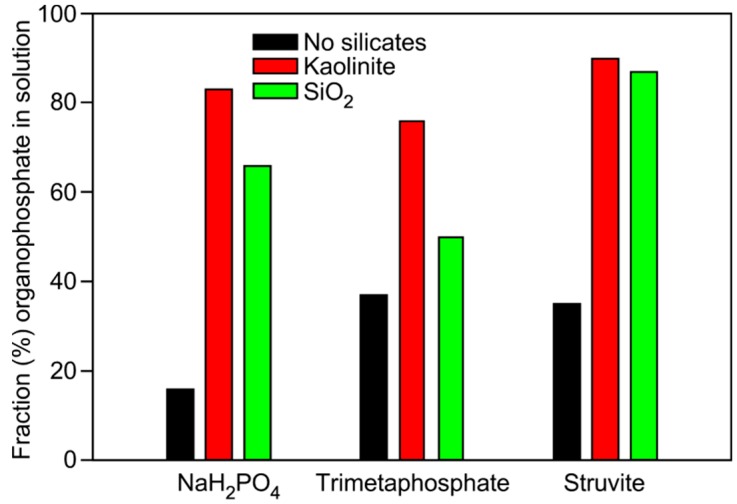
Bar graph showing the total yield of organophosphate from reactions performed in formamide. Differences in phosphorylation for reactions in formamide with quartz sand, kaolinite, or no mineral catalyst, and with the different phosphorus sources NaH_2_PO_4_, TMP, and struvite. Formamide appears to be a better phosphorylating medium than the glycerol DES with the highest yields coming from using struvite in the presence of kaolinite. Errors are about 1%, based on repetition of integration of NMR peaks.

**Table 1 life-07-00029-t001:** Yields ^a^ of organophosphates (%) in glycerol-choline chloride eutectic solvent (DES).

Phosphate Source	Mineral Added	Glycerol-1-phosphate	Choline-phosphate	Glycerol-2-phosphate	Glycerol-1,2-cyclic phosphate	Total
NaH_2_PO_4_	None	0	0	0	0	0
NaH_2_PO_4_	Kaolinite	8	3	1	0	12
NaH_2_PO_4_	Qtz Sand	7	0	0	0	7
TMP	None	11	0	4	0	15
TMP	Kaolinite	35	10	12	5	76
TMP	Qtz Sand	18	3	3	1	25
Struvite	None	8	5	2	0	15
Struvite	Kaolinite	21	4	3	0	28
Struvite	Qtz Sand	12	10	0	0	22

^a^ The yields were calculated on the basis of the total phosphorus dissolved into the solvent and by the peak integration method, coupled to semi-quantitative concentration estimation using signal to noise ratios [1].

**Table 2 life-07-00029-t002:** Yields ^a^ of organophosphates (%) in formamide.

Phosphate	Mineral Added	Glycerol-1-phosphate	Glycerol-2-phosphate	Glycerol-1,2-cyclic phosphate	Total
NaH_2_PO_4_	None	12	4	0	16
NaH_2_PO_4_	Kaolinite	70	13	0	83
NaH_2_PO_4_	Qtz Sand	56	10	0	66
TMP	None	35	2	0	37
TMP	Kaolinite	76	0	0	76
TMP	Qtz Sand	38	11	1	50
Struvite	None	30	5	0	35
Struvite	Kaolinite	78	11	1	90
Struvite	Qtz Sand	75	9	2	87

^a^ The yields were calculated on the basis of the total phosphorus dissolved into the solvent and by the peak integration method, coupled to semi-quantitative concentration estimation using signal to noise ratios [1].

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
