# Peer review of "Silicate-Promoted Phosphorylation of Glycerol in Non-Aqueous Solvents: A Prebiotically Plausible Route to Organophosphates"

_life, 2017, doi:10.3390/life7030029_

Round 1

Reviewer 1 Report

Major comments

The experiments are designed to use identical amount of glycerol and identical weight of phosphate. So, molar amounts of phosphate added to the TMP experiment and NaH2PO4 experiment are different. Further, if these phosphates do not dissolve completely, the molar ratio of reactants (glycerol/dissolved phosphate) are complicated and differs in different experiments. The glycerol/dissolved phosphate ratio should affect the yields. In particular in the present manuscript, the yields are shown based on the ratio of phosphorus distributed in products among of the dissolved phosphate. The glycerol/dissolved phosphate ratio may affect significantly to the yield number, if the solubility of TMP and Struvite is significantly low. I hope this point is discussed is revised manuscript.

This would be the next phase of this work but the reaction rate of phosphorylation differs in different conditions as shown in Schoffstall’s paper and Furukawa’s paper in the reference list. All of the present data in this manuscript is taken in different conditions (i.e., silicate additives and solvents) under the same temperate and duration. So, the yields would differ when the incubation continued longer or shorter. This point should be mentioned in the revised manuscript.

Minor comments

L93: Most of all commercial kaolinite contains other minerals such as quartz and alunite. The author should provide more information on kaolinite they used. Also, it’s helpful to be mentioned on particle size of quartz sand.

L103-104 “in organic” -> “inorganic”

L103: The author should show whether inorganic phosphates are completely dissolved in the experimental solvents or not. If not, it’s better to show how much phosphate was dissolved in solvents in the beginning and end of the experiments.

Caption of Fig. 2: GES should be defined when it appears first.

Table 1: The column name and numbers in “total” in table 1 is strange. Maybe they are just shifted. Also, the number in NaH2PO4 should be changed to lower case.

Figure 2(C) and L132-135: Why TMP was hydrolyzed in GES and formamide? What is the source of water? Water released by the phosphorylation of glycerol? In that case, the activity of water in TMP experiments are lower than others.

Figure 3 and 4: The unit of the vertical axis would be %. Also, “total organophosphate in dissolved phosphate” is better.

Author Response

The experiments are designed to use identical amount of glycerol and identical weight of phosphate. So, molar amounts of phosphate added to the TMP experiment and NaH2PO4 experiment are different. Further, if these phosphates do not dissolve completely, the molar ratio of reactants (glycerol/dissolved phosphate) are complicated and differs in different experiments. The glycerol/dissolved phosphate ratio should affect the yields. In particular in the present manuscript, the yields are shown based on the ratio of phosphorus distributed in products among of the dissolved phosphate. The glycerol/dissolved phosphate ratio may affect significantly to the yield number, if the solubility of TMP and Struvite is significantly low. I hope this point is discussed is revised manuscript.

>>This is a quite valid point.  We have added the following paragraph:

“Sodium phosphate and sodium trimetaphosphate both dissolved to completion in formamide.  Struvite was less soluble, dissolving at about 1 g/L, which is still higher than its solubility in water [39].  All salts were less soluble in the glycerol-choline chloride eutectic, with TMP most soluble at 0.2 g/L, sodium phosphate 0.02 g/L, and struvite 0.01 g/L.  In all cases, dissolved phosphate was the limiting reagent.“

This would be the next phase of this work but the reaction rate of phosphorylation differs in different conditions as shown in Schoffstall’s paper and Furukawa’s paper in the reference list. All of the present data in this manuscript is taken in different conditions (i.e., silicate additives and solvents) under the same temperate and duration. So, the yields would differ when the incubation continued longer or shorter. This point should be mentioned in the revised manuscript.

>> We have added a paragraph to this effect after table 2: "The experiments performed here were all measured at one week.  It is likely that some of these reactions will provide different yields upon longer or shorter incubation times, as suggested by prior work [21, 22, 24].  The rate of phosphorylation reaction may be slow, especially those reactions contingent of dissolution of phosphate minerals (e.g., struvite), hence completion may be longer than the one week investigated here."

Minor comments

L93: Most of all commercial kaolinite contains other minerals such as quartz and alunite. The author should provide more information on kaolinite they used. Also, it’s helpful to be mentioned on particle size of quartz sand.

>>Quartz sand is 200-800 um grain size (added).  Information on the kaolin is less easy to find from the supplier, but XRD shows no additional minerals (>5%).

L103-104 “in organic” -> “inorganic”

>>Changed

L103: The author should show whether inorganic phosphates are completely dissolved in the experimental solvents or not. If not, it’s better to show how much phosphate was dissolved in solvents in the beginning and end of the experiments.

>> See above under major comments. 

Caption of Fig. 2: GES should be defined when it appears first.

>>Changed to glycerol deep eutectic solvent.

Table 1: The column name and numbers in “total” in table 1 is strange. Maybe they are just shifted. Also, the number in NaH2PO4 should be changed to lower case.

>> Fixed

Figure 2(C) and L132-135: Why TMP was hydrolyzed in GES and formamide? What is the source of water? Water released by the phosphorylation of glycerol? In that case, the activity of water in TMP experiments are lower than others.

>> In this case, the water might either be intrinsic to the glycerol or formamide (both which may have up to a half percent H2O), or from the dehydration reaction as the reviewer proposed, or from trace humidity from doing these experiments in Florida.  We agree that TMP is acting as a drying agent with lower H2O activity.  We have added the following text: “a hydrolysis product of TMP, with water promoting this hydrolysis likely coming from the phosphorylation reaction or present initially in the solvent (<0.5 wt.%) or from ambient humid conditions.”

Figure 3 and 4: The unit of the vertical axis would be %. Also, “total organophosphate  in dissolved phosphate” is better.

>>Changed

Reviewer 2 Report

The authors describe the synthesis of mixtures of glycerol mono-ester phosphate derivatives by reaction of glycerol with different forms of inorganic phosphate, comprising a mineral struvite, in the presence of formamide or, in alternative, of an eutectic solvent consisting of choline chloride and glycerol. The reaction was also studied in the presence of silicates such as quartz sand and kaolinite clay. The authors clearly suggest the possible relevance of these results in the prebiotic scenario, in connection with the origin of membranes or of other important biomolecules. As stated by the authors in the introduction, “for over 40 years, formamide has been examined as both a reactant and solvent for the formation of biomolecules under model prebiotic conditions”. Unfortunately, the references they cited about (ref. 16. Yamada, H. et. al.  Chem. Pharm. Bull., 1972, and ref. 17 Gull, M. Challenges 2014) are not the most representative. Yamada only described the synthesis of the simple purine from formamide, and purine alone is not a useful molecule for the origin of nucleic acids. The study of Gull only marginally describes the prebiotic chemistry of formamide. Many and more significant articles should be mentioned about the importance of formamide in this field. On the other hand, the key point is: The results described by the authors are new enough to increase the knowledge of the field, and are they a plausible prebiotic model? Analyzing the literature, and considering the reported experimental details, the answer is negative. The phosphorylation of primary and secondary alcoholic moieties in sugars has been previously (and largely) reported as in refs 21 -24 simply by heating the substrate in formamide in the presence of inorganic phosphate or mineral containing phosphates at room pressure. In these latter cases, formamide was able to solubilize directly and activate the phosphate for phosphorylation. Instead, as stated by the authors, choline chloride requires that “the reaction vials were sealed tightly and heated with constant stirring at 85 oC for one week”. Are these prebiotic conditions? Moreover, in order to highlight the prebiotic importance of choline chloride, the authors reported that “choline has been prepared under prebiotic conditions [39-41]”. Unfortunately, none of the citations mentioned shows what the authors say. Singh (ref 39) only reports that the choline chloride/glycerol eutectic solvent is produced by reaction between commercially available choline and urea. Austin (ref 40) o describe the synthesis of a pyridoxal derivative, while different low molecular weight compounds are reported in the study of Miller (ref 41), but any prebiotic synthesis of choline chloride is described.  The work is poorly written, e.g. the figures are devoid of the indication of the experimental error, which is also missing in the description of the methods in the experimental part. Any experimental evidence is given by the authors about the possible role of quartz sand and kaolinite clay in the reaction mechanism, despite the authors highlighted that “To the best of our knowledge, this is the first study showing a two-mineral, one-pot reaction produces higher yields than single mineral reactions, suggesting either some interplay between the mineral reactivity, or catalytic activity promoting the reaction”. Finally, the sentence “the heating of the reaction mixtures here for one week in the presence of silicates also promotes the formation of glycerol cyclic phosphate (pag 7, line 177), tempted the reader to believe that this represents the current experimental, but the synthesis of cyclic phosphates was previously reported in ref. 23.  In my opinion the manuscript is not adequate to be published on Life.

Author Response

The authors describe the synthesis of mixtures of glycerol mono-ester phosphate derivatives by reaction of glycerol with different forms of inorganic phosphate, comprising a mineral struvite, in the presence of formamide or, in alternative, of an eutectic solvent consisting of choline chloride and glycerol. The reaction was also studied in the presence of silicates such as quartz sand and kaolinite clay. The authors clearly suggest the possible relevance of these results in the prebiotic scenario, in connection with the origin of membranes or of other important biomolecules. As stated by the authors in the introduction, “for over 40 years, formamide has been examined as both a reactant and solvent for the formation of biomolecules under model prebiotic conditions”. Unfortunately, the references they cited about (ref. 16. Yamada, H. et. al.  Chem. Pharm. Bull., 1972, and ref. 17 Gull, M. Challenges 2014) are not the most representative. Yamada only described the synthesis of the simple purine from formamide, and purine alone is not a useful molecule for the origin of nucleic acids. The study of Gull only marginally describes the prebiotic chemistry of formamide. Many and more significant articles should be mentioned about the importance of formamide in this field.

>>We have modified the references to formamide to include 3 new ones from the Di Mauro group, replacing Yamada et al. Furthermore, references starting from 16, up to ref 27, we have tried to reference almost all the important aspects of formamide such as its prebiotic relevance, its availability, and its application in prebiotic chemistry as a solvent or a precursor.

On the other hand, the key point is: The results described by the authors are new enough to increase the knowledge of the field, and are they a plausible prebiotic model?  Analyzing the literature, and considering the reported experimental details, the answer is negative. The phosphorylation of primary and secondary alcoholic moieties in sugars has been previously (and largely) reported as in refs 21 -24 simply by heating the substrate in formamide in the presence of inorganic phosphate or mineral containing phosphates at room pressure. In these latter cases, formamide was able to solubilize directly and activate the phosphate for phosphorylation. Instead, as stated by the authors, choline chloride requires that “the reaction vials were sealed tightly and heated with constant stirring at 85 oC for one week”. Are these prebiotic conditions?

>> The sealing is not necessary to the reaction, but was done to limit other factors, such as changing ambient humidity common to Florida.  If water is allowed to escape from the vials, we expect yields to be higher.

Moreover, in order to highlight the prebiotic importance of choline chloride, the authors reported that “choline has been prepared under prebiotic conditions [39-41]”. Unfortunately, none of the citations mentioned shows what the authors say. Singh (ref 39) only reports that the choline chloride/glycerol eutectic solvent is produced by reaction between commercially available choline and urea.  Austin (ref 40) o describe the synthesis of a pyridoxal derivative, while different low molecular weight compounds are reported in the study of Miller (ref 41), but any prebiotic synthesis of choline chloride is described. 

>> The reviewer is correct that choline chloride has not actually been identified through a prebiotic synthesis (with respect to the references we cite).  We have modified this to read, “Choline may be accessible under prebiotic conditions [39-41],”

We do not feel this impacts the paper significantly because 1) several other prebiotic experiments have assumed the presence of choline, including Deamer and Barchfeld (JME 1982), Mar et al. (OLEB 1987), and Aylward (Proc WSEAS 2011).  And 2) the presence of choline making a eutectic primarily lowers the freezing point of glycerol to below 18°C.  Given that these experiments operate in excess of this temperature, it may be reasonable to assume that the results would not be impacted on removal of choline chloride as there are lab-formed glycol-phosphate polymers, likely similar to glycerol-polymers (Pretula et al. J Polymer Sci A 2008).

The work is poorly written, e.g. the figures are devoid of the indication of the experimental error, which is also missing in the description of the methods in the experimental part.

>> Errors are about 1%, based on repetition of integration of NMR peaks.  This has been added to the figure captions of 3 & 4. 

Any experimental evidence is given by the authors about the possible role of quartz sand and kaolinite clay in the reaction mechanism, despite the authors highlighted that “To the best of our knowledge, this is the first study showing a two-mineral, one-pot reaction produces higher yields than single mineral reactions, suggesting either some interplay between the mineral reactivity, or catalytic activity promoting the reaction”.

>> We’ve declined to speculate here, as sometimes the mineral serves to catalyze the reaction merely by providing more surface area, other times it takes a more active role and binds the reactants, and still other times it actively participates in the reaction. 

Our presented research work primarily focuses on the phosphorylation reactions i.e. studying the comparison of the phosphorylation rates by different phosphorylating agents such as phosphate mineral (struvite), condensed phosphate (TMP), and phosphate salt (NaH2PO4) in different non-aqueous solvents, some catalysts, and using some non-aqueous solvents. In near future, we aim to perform exhaustive study on the mechanism of silicate catalysis but at this stage we only suggest that organic substrates could bind to silicates and thus get stabilized to stay long enough at elevated temperature to get phosphorylated but this has to be studied in future work.

Finally, the sentence “the heating of the reaction mixtures here for one week in the presence of silicates also promotes the formation of glycerol cyclic phosphate (pag 7, line 177), tempted the reader to believe that this represents the current experimental, but the synthesis of cyclic phosphates was previously reported in ref. 23.

>> The cyclic glycerol phosphate is part of this current work, though other cyclic phosphates have certainly been demonstrated by prebiotic phosphorylation experiments (including Schoffstall 1976). It is omitted from the NMR spectra merely because they occur at ~19 ppm, and their inclusion expands the x-axis such that peak separation is lost from the monoester region, and because they’re not the most abundant species.

Reviewer 3 Report

I have read the article easily. It is a clear piece of work that merits publication. It is short, clear, precise and well described. The results and finding are novel and interesting and they are a new laboratory support to the formamide route to origins of life. The only comment is about the plausibility of glicerol as a prebiotic chemistry but I like the tone of prudence about this and other items in the paper. 

Author Response

I have read the article easily. It is a clear piece of work that merits publication. It is short, clear, precise and well described. The results and finding are novel and interesting and they are a new laboratory support to the formamide route to origins of life. The only comment is about the plausibility of glicerol as a prebiotic chemistry but I like the tone of prudence about this and other items in the paper.

>> We agree with the reviewer about the prebiotic relevance of glycerol, and have given Epps et al. (1979) as one more relevant reference.

Round 2

Reviewer 2 Report

Unfortunately I can not modify my first comment about this article, rejected. As  the same authors emphasize in their response, the chemical system based  on choline has no evidence of having a prebiotic origin.The claim of the authors that "several other prebiotic experiments have assumed the  presence of choline"  adds no significance to a  very basic consideration for the seriousness of prebiotic chemistry. Since  there is no direct evidence of what has actually happened, the only  possibility is to rely on experimental data and not on the opinions of  some of the researchers, as quoted. Furthermore,  it is not understandable  the relationships between the criticism on the fact that the reactions are conducted in a closed  vial and the authors' statement about "changing ambient humidity common  to Florida"!?! Life emerged in Florida ??